# Association of Periodontal Disease with the Occurrence of Unruptured Cerebral Aneurysm among Adults in Korea: A Nationwide Population-Based Cohort Study

**DOI:** 10.3390/medicina57090910

**Published:** 2021-08-30

**Authors:** Ho-Geol Woo, Yoon-Kyung Chang, Ji-Sung Lee, Tae-Jin Song

**Affiliations:** 1Department of Neurology, Kyung Hee University College of Medicine, Seoul 02447, Korea; nr85plasma@naver.com; 2Department of Neurology, Mokdong Hospital, Ewha Womans University College of Medicine, Seoul 07985, Korea; tin1207@nate.com; 3Clinical Research Center, Asan Medical Center, Seoul 05505, Korea; totoro96a@gmail.com; 4Department of Neurology, Seoul Hospital, Ewha Womans University College of Medicine, Seoul 07804, Korea

**Keywords:** oral hygiene, periodontal disease, unruptured cerebral aneurysm, longitudinal study

## Abstract

*Background and Objectives:* Cerebral aneurysms can cause disability or death during rupture, but information on the etiology of cerebral aneurysms is currently lacking. Periodontal disease causes both systemic inflammation and local inflammation of the oral cavity. Systemic inflammation is a major cause of cerebral aneurysms. The aim of our study was to determine whether the presence of periodontal disease is related to the occurrence of unruptured cerebral aneurysms in a nationwide population-based cohort. *Materials and Methods:* We analyzed data on demographics, previous medical history, and laboratory test results of 209,620 participants from the Korean National Health Insurance System-Health Screening Cohort. The presence of periodontal disease and oral hygiene parameters, including the number of lost teeth, tooth brushing frequency per day, dental visits for any reason, and expert teeth scaling, were investigated. The occurrences of unruptured cerebral aneurysms (I67.1) were defined according to the International Statistical Classification of Diseases Related Health Problems-10. *Results:* The mean age of the participants was 53.7 ± 8.7 years, and 59.4% were male. Periodontal disease was found in 20.9% of the participants. A total of 2160 (1.0%) cases of unruptured cerebral aneurysms developed after 10.3 years of median follow up. In multivariate analysis, the presence of periodontal disease was significantly associated with an increased risk of unruptured cerebral aneurysms (hazard ratio: 1.21, 95% confidence interval: 1.09–1.34, *p* < 0.001). *Conclusion:* The presence of periodontal disease could be associated with the occurrence of unruptured cerebral aneurysms. It should be noted that when periodontal diseases are present, the risk of aneurysms is increased in the future.

## 1. Introduction

An unruptured cerebral aneurysm is characterized by abnormal dilatation of the vessel wall in the cerebral arteries and affects 3–5% of the adult population [1]. Approximately 1% of unruptured cerebral aneurysms rupture per year, which can cause subarachnoid hemorrhage that may, in turn, result in severe neurological deficits and sudden death [2]. Therefore, it is important to know the causes of unruptured cerebral aneurysms. The factors known to be associated with an unruptured cerebral aneurysm are hemodynamic stress, uncontrolled hypertension, atherosclerosis, smoking, and genetic conditions [3], but more information on modifiable risk factors is still needed.

Periodontal disease, including periodontitis and gingivitis, is related to oral hygiene [4]. A previous case series reported that periodontitis is frequently observed in patients with an unruptured cerebral aneurysm [5]. One study showed that periodontitis and gingival bleeding are related to an increased risk of unruptured cerebral aneurysm formation [6]. Moreover, oral bacterial DNA has been found in the walls of unruptured cerebral aneurysms (71%), suggesting that the oral bacteria causing periodontal disease may also play a part in these aneurysms [7]. However, the relationship between the occurrence of unruptured cerebral aneurysms and oral hygiene parameters has rarely been noted. Moreover, longitudinal studies investigating the relationship between the occurrence of unruptured cerebral aneurysms and oral hygiene parameters and periodontal disease are rare. Our study hypothesized that periodontal disease and the parameters for oral hygiene would be positively correlated to unruptured cerebral aneurysms in the data from Korea’s health insurance system health checkup.

## 2. Materials and Methods

### 2.1. Data Sources

The National Health Insurance System (NHIS) is Korea’s health insurance system, which includes data on demographics, methods of diagnosis, and treatment. Health screening is offered every two years for NHIS members, and oral health screening is offered for NHIS members aged 40 years and above [8]. The NHIS offers random sampling information of the health screening of 50 million Koreans from 2002 to 2015 for researchers [9].

### 2.2. Participants

We enrolled study participants aged between 40 and 79 years from the NHIS-National Health Screening Cohort (NHIS-HEALS) from 2003 to 2006, which includes data about not only the weight, height, socioeconomic status of participants, their lifestyle questionnaires, and their laboratory test results, but also their periodontal disease and oral hygiene parameters [10]. A more than one-year washout period was established to rule out participants with unruptured cerebral aneurysms at the beginning of the study. The date of the oral health check-up was defined as the index date for the analysis. When there were serial measurements for each variable, the latest value during the period between 2003 and 2006 was utilized for the analysis. We eliminated participants with missing data. We included 209,620 participants in the analysis using the following criteria (Figure 1).

### 2.3. Study Variables and Definitions

The definition of risk factors and concomitant disease is recounted in the Appendix A. The presence of periodontal disease was defined according to previous studies, when dentists claimed relevant ICD-10 codes (acute periodontitis (K052), chronic periodontitis (K053), periodontosis (K054), other periodontal diseases (K055), and unspecified periodontal disease (K056)) more than twice or when participants went to the dentist for treatment (health claim codes: U1010, U1020, U1051-1052, U1071-1072, and U1081-1083) with the diagnostic code of periodontal disease (K052-056) in the preceding year [10,11,12,13]. Dentists confirmed the number of lost teeth, which was classified as none, one to five, and six or more with tertile value regardless of the causative disease. Oral hygiene behavior (tooth brushing frequency per day, dental visits for any reason, and expert teeth scaling) was collected from question investigation [10]. Tooth brushing frequency per day was classified as none or once, twice, and thrice or more times. Dental visits for any reason and expert teeth scaling were classified as never or at least once per year. If the participant had a dental visit or received expert teeth scaling once every two years, they were categorized as part of the never group. Prior studies have proved high diagnostic acuity in the identification of comorbidities using the ICD-10 code, which is based on the health claims data in the NHIS [14,15,16].

The primary outcome was the occurrence of an unruptured cerebral aneurysm, which was considered as new registration of ICD-10 code I67.1 claimed more than twice per year, with brain imaging using computed tomography or magnetic resonance imaging obtained at the time of diagnosis.

### 2.4. Statistical Analysis

Univariate analysis was analyzed with the *t*-test and the chi-square test, as appropriate. In addition, we compared the demographics of the participants included and excluded using standardized differences; greater than 0.1 was regarded as significant. Kaplan–Meier survival curves were performed to estimate the association between the occurrence of unruptured cerebral aneurysms and oral hygiene parameters and the presence of periodontal disease. Statistical differences were analyzed using the log-rank test. We evaluated the relationship between the incidence of unruptured cerebral aneurysms and oral hygiene parameters using the Cox proportional hazards model. Hazard ratios (HRs) and 95% confidence intervals (CIs) were calculated. Multivariate Cox regression analyses were used to test the association between each oral hygiene parameter and the occurrence of an unruptured cerebral aneurysm after adjusting for demographics, socioeconomic status, regular physical activities, body mass index, alcohol intake, smoking status, and comorbid diseases in model 1; adjusting for the variables of model 1 along with systolic blood pressure, fasting blood sugar, liver panel tests, and proteinuria in model 2; and adjusting for the variables of model 2 and the presence of periodontal disease and parameters for oral hygiene in model 3. The HRs were analyzed for the presence of periodontal disease and the parameters for oral hygiene after adjusting for the above-mentioned factors in models 1 and 2. To evaluate the trends in the HRs regarding the frequency of tooth brushing and the number of lost teeth, a *p*-value for the trend was assessed. There was no multicollinearity for the risk of cerebral aneurysms associated with the oral hygiene parameters in the multivariate analysis (variance inflation factors were less than 5.0; Appendix A). Statistical analyses were conducted using SAS software (version 9.2, SAS Institute, Cary, NC, USA). A *p*-value of <0.05 was regarded as statistically significant.

## 3. Results

When comparing the demographic data between the participants included and excluded in this study, a higher socioeconomic status was more frequent in the included participants (Appendix A). The mean age of the participants was 53.7 ± 8.7 years and 59.4% were male. The percentage of participants with periodontal disease was 20.9%. Participants with none, one to five, and six or more lost teeth accounted for 75.4%, 20.7%, and 3.9%, respectively. The rates of brushing teeth—none or once, twice, and thrice or more times a day—were 14.3%, 43.7%, and 42.0%, respectively. The percentage of participants who underwent expert teeth scaling was 25.9% (Table 1). Details of the laboratory findings are described in Table 1. Participants with periodontal disease had poor oral hygiene behaviors and a greater number of lost teeth compared to participants without periodontal disease (Table 1).

During a median follow-up of 10.3 years (interquartile range, 9.5–11.6), 2160 (1.0%) participants developed unruptured cerebral aneurysms. Moreover, among the 2160 participants that developed unruptured cerebral aneurysms, 500 (23.1%) had periodontal disease, and among the 43,806 participants with periodontal disease, 500 (1.1%) participants had unruptured cerebral aneurysms. The Kaplan–Meier survival curves showed that the occurrence of unruptured cerebral aneurysms depends on the presence of periodontal disease (*p* < 0.001) and the number of lost teeth (*p* = 0.001), but not tooth brushing frequency per day, dental visits for any reason, or expert teeth scaling (Figure 2).

The presence of periodontal disease was positively related to the occurrence of unruptured cerebral aneurysms in model 1 (HR: 1.23, 95% CI: 1.11–1.36, *p* < 0.001), model 2 (HR: 1.23, 95% CI: 1.11–1.36, *p* < 0.001), and model 3 (HR: 1.21, 95% CI: 1.09–1.34, *p* < 0.001) in the multivariate analysis (Table 2). In contrast, the number of lost teeth did not show a significant association with the future occurrence of unruptured cerebral aneurysms after adjustment in models 1–3 (Table 2).

The subgroup analysis demonstrated no statistically significant interaction between periodontal disease and unruptured cerebral aneurysms related to age, sex, or smoking status.

## 4. Discussion

Our present study verified that the presence of periodontal disease positively correlates with the occurrence of unruptured cerebral aneurysms after adjusting for confounding variables. In a previous study, periodontitis and gingival bleeding were related to an increased occurrence of intracranial aneurysms, and severe periodontitis was commonly observed in patients with intracranial aneurysms during preoperative dental examination [5,6]. In addition, poor oral hygiene is a predisposing factor for the formation of mycotic intracranial aneurysms [17]. Therefore, our results are consistent with those of previous studies and provide additional knowledge about the relationship between the presence of periodontal disease and the occurrence of cerebral aneurysms in the general population in a longitudinal study setting.

Our study did not show a relationship between the number of lost teeth and the occurrence of unruptured cerebral aneurysms, although there are few reports of an association between the number of lost teeth and cerebral aneurysms. For example, a previous study showed that the number of lost teeth was related to the presence of an aortic aneurysms [18]. Our findings suggest that the number of lost teeth itself may not increase the risk of developing cerebral aneurysms. However, there may be differences in the study design, method, the subjects, and the sampling method; therefore, further studies are necessary to prove these relationships.

Although tooth brushing is an easy procedure, it is an important health behavior that can improve oral hygiene. In previous studies, frequent tooth brushing was shown to be dose-dependently associated with a decreased occurrence of atrial fibrillation, heart failure, stroke, and new-onset diabetes mellitus [11,19,20]. In other words, improved oral hygiene behavior decreases the risk of a variety of cardiovascular diseases. However, in our study, frequent tooth brushing was not related to the occurrence of unruptured cerebral aneurysms. This is the reason why the occurrence of unruptured cerebral aneurysms might not be controlled only by frequent tooth brushing.

The current understanding of unruptured cerebral aneurysms is that they are formed as a result of blood flow-induced, inflammation-mediated wall remodeling of cerebral arteries [21]. However, since all cerebral arteries exposed to hemodynamic stress do not form cerebral aneurysms, additional factors have been suggested [21]. For example, a previous review suggested that chronic inflammation is associated with the formation and growth of cerebral aneurysms [22]. Poor oral hygiene and periodontal disease can provoke the invasion of periodontal pathogens into the systemic circulation, causing a systemic inflammatory response. Systemic immune processes are known to facilitate the occurrence of unruptured cerebral aneurysms [23]. Moreover, a recent animal study showed that temporary ablation of the gut microbiome inhibits the formation of unruptured cerebral aneurysms by modulating inflammation [24]. Although our results could not conclude causative pathophysiology regarding the association of periodontal diseases with the occurrence of unruptured cerebral aneurysms, the results of previous studies support our findings that the presence of periodontal disease may increase the risk of unruptured cerebral aneurysms.

The current study has some limitations. First, the results cannot be generalized to populations of different ethnicities, because the dataset only included the Korean population. Further study demonstrating whether or not there are differences depending on race is needed. Second, because the cohort dataset does not provide all information, a few confounding variables, including inflammatory blood biomarkers, educational status, and marital status, were not included in our analysis. Third, we used a self-reported survey about oral hygiene parameters, in which recall bias that may have occurred. Fourth, because we collected data about oral hygiene parameters and various variables at the index date, factors varying with time could not be considered. Fifth, in our dataset, various severities of periodontal disease could not be analyzed. Therefore, further study considering the severity of periodontal disease and the number of years of history is needed. Sixth, the definition of periodontal disease used in our study is based on the ICD-10 code, which does not reflect the recently used classification criteria and case definition for periodontal disease [25]. Therefore, further study using the recently updated periodontal disease classification is needed. Seventh, the definition of an unruptured cerebral aneurysm was based on the ICD-10 code using health claim data. Unruptured cerebral aneurysms are usually asymptomatic and can be diagnosed using expensive imaging studies. Although we applied a one-year washout period before the index date, the possibility of a hidden unruptured cerebral aneurysm at baseline still remains, and under-reporting might have been possible. However, some previous reports used a definition for the presence of unruptured cerebral aneurysms similar to the method used in our study [26,27]. Nevertheless, there was a paucity of events with a low aneurysm rate, which decreased the significance of our results. Eighth, although the demographic data did not show a significant difference, the exclusion of many participants, particularly those who did not undergo dental examination, may have led to a selection bias. Ninth, our dataset did not include confounding variables, including the use of interdental devices, sonic brushes with vertical movement, and oral irrigators. As the use of sonic brushes, interdental devices, and oral irrigators is related to plaque control and a lower incidence of periodontal disease, caries, and missing teeth, future studies considering variables such as the use these devices are needed [28,29].

## 5. Conclusions

The current study demonstrated that the presence of periodontal disease is associated with an increased risk of occurrence of cerebral aneurysms. It should be noted that when periodontal diseases are present, the risk of aneurysms increases in the future.

## Figures and Tables

**Figure 1 medicina-57-00910-f001:**
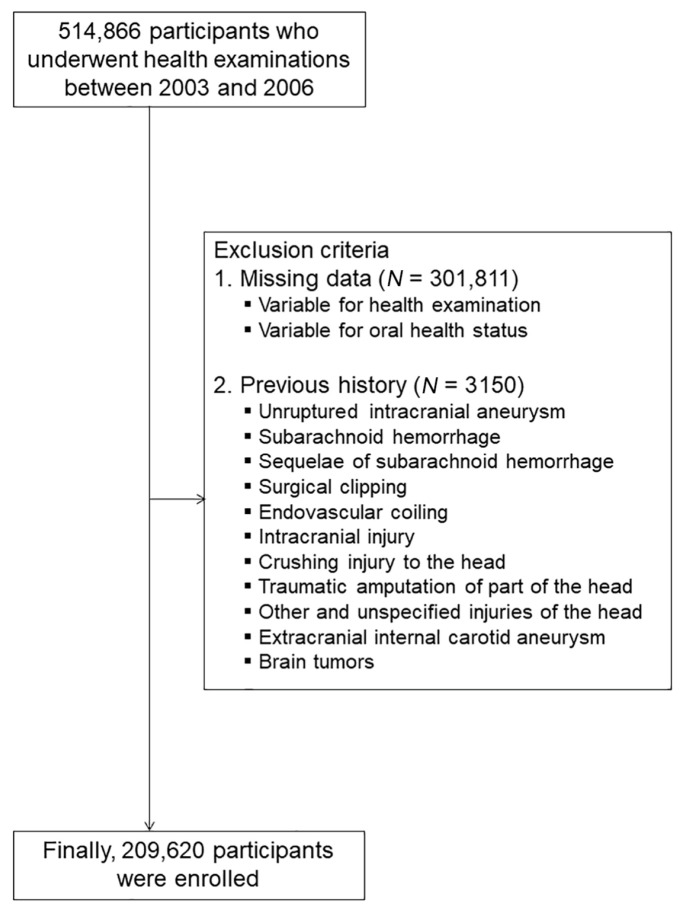
Flowchart of the study participants.

**Figure 2 medicina-57-00910-f002:**
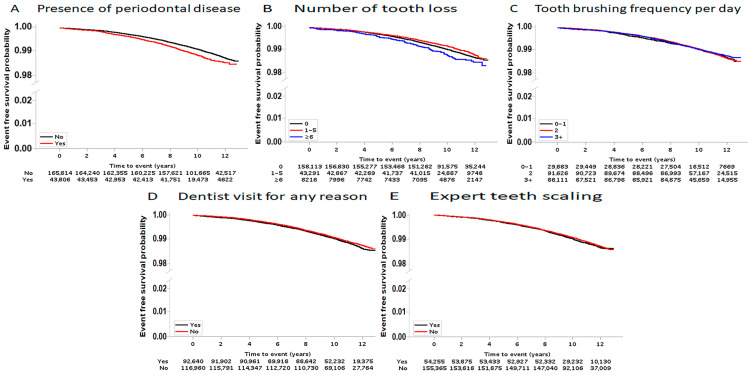
Kaplan–Meier survival curves associated with oral hygiene parameters and the presence of periodontal disease for the risk of cerebral aneurysm development. The Kaplan–Meier curves shows that the risk of cerebral aneurysm development depends on the presence of periodontal disease (**A**) (*p* < 0.001) and the number of lost teeth (**B**) (*p* = 0.001), but not tooth brushing frequency per day (**C**) (*p* = 0.846), dental visits for any reason (**D**) (*p* = 0.125), or expert teeth scaling (**E**) (*p* = 0.337).

**Table 1 medicina-57-00910-t001:** Baseline characteristics between participants with and without periodontal disease.

Characteristics	Total	Periodontal Disease		*p*-Value
		No	Yes	
Number of participants	209,620	165,814	43,806	
Age (years)	53.7 ± 8.7	53.5 ± 8.8	54.3 ± 8.2	<0.001
Male sex	124,496 (59.4)	95,102 (57.4)	29,394 (67.1)	<0.001
Socioeconomic status				<0.001
Fifth quintile (highest)	83,828 (40.0)	63,891 (38.5)	19,937 (45.5)	
Fourth quintile	41,931 (20.0)	33,319 (20.1)	8612 (19.7)	
Third quintile	29,697 (14.2)	24,060 (14.5)	5637 (12.9)	
Second quintile	26,563 (12.7)	21,855 (13.2)	4708 (10.7)	
First quintile (lowest)	27,171 (13.0)	22,310 (13.5)	4861 (11.1)	
Covered by medical aid	430 (0.2)	379 (0.2)	51 (0.1)	
Regular physical activity	21,255 (10.1)	16,476 (9.9)	4779 (10.9)	<0.001
Alcohol intake	92,661 (44.2)	71,754 (43.3)	20,907 (47.7)	<0.001
Body mass index (kg/m^2^)	24.0 ± 2.9	23.9 ± 2.9	24.1 ± 2.8	<0.001
Systolic blood pressure (mmHg)	125.8 ± 16.6	125.8 ± 16.8	125.8 ± 16.0	0.876
Diastolic blood pressure (mmHg)	78.6 ± 10.7	78.6 ± 10.8	78.6 ± 10.5	0.522
Comorbidities				
Hypertension	77,617 (37.0)	60,610 (36.6)	17,007 (38.8)	<0.001
Diabetes mellitus	21,300 (10.2)	15,575 (9.4)	5725 (13.1)	<0.001
Dyslipidemia	38,373 (18.3)	29,483 (17.8)	8890 (20.3)	<0.001
Current smoker	44,421 (21.2)	33,726 (20.3)	10,695 (24.4)	<0.001
Renal disease	586 (0.3)	436 (0.3)	150 (0.3)	0.005
History of malignancy	25,345 (12.1)	18,768 (11.3)	6577 (15.0)	<0.001
Laboratory findings				
Total cholesterol (mg/dL)	197.8 ± 36.3	197.8 ± 36.3	197.7 ± 36.2	0.557
Fasting blood glucose level (mg/dL)	97.8 ± 27.4	97.3 ± 26.9	99.7 ± 29.0	<0.001
Aspartate aminotransferase (U/L)	26.5 ± 16.1	26.5 ± 16.2	26.6 ± 15.5	0.299
Alanine aminotransferase (U/L)	25.6 ± 20.1	25.4 ± 20.0	26.2 ± 20.3	<0.001
Gamma-glutamyl transferase (U/L)	38.5 ± 54.5	38.1 ± 55.3	40.0 ± 51.4	<0.001
Proteinuria (≥1+ in dip stick test)	6839 (3.3)	5315 (3.2)	1524 (3.5)	0.004
Oral health status				
Presence of periodontal diseases	43,806 (20.9)	N/A	N/A	
Number of lost teeth				<0.001
0	158,113 (75.4)	127,105 (76.7)	31,008 (70.8)	
1–5	43,291 (20.7)	32,208 (19.4)	11,083 (25.3)	
≥6	8216 (3.9)	6501 (3.9)	1715 (3.9)	
Oral hygiene care				
Tooth brushing frequency per day				<0.001
0–1	29,883 (14.3)	24,312 (14.7)	5571 (12.7)	
2	91,626 (43.7)	74,067 (44.7)	17,559 (40.1)	
≥3	88,111 (42.0)	67,435 (40.7)	20,676 (47.2)	
Dental visits for any reason	92,640 (44.2)	63,675 (38.4)	28,965 (66.1)	<0.001
Expert teeth scaling	54,255 (25.9)	35,711 (21.5)	18,544 (42.3)	<0.001

Notes: *p*-Value by Student’s *t*-test and the chi-square test. Data are expressed as mean ± standard deviation or *n* (%). N/A, not applicable.

**Table 2 medicina-57-00910-t002:** Risks of cerebral aneurysms associated with oral hygiene parameters.

	Number of Patients	Number of Events	Event Rate (95% CI)	Unadjusted Model		Multivariate Adjusted Model (1)		Multivariate Adjusted Model (2)		Multivariate Adjusted Model (3)	
				HR (95% CI)	*p*-Value	HR (95% CI)	*p*-Value	HR (95% CI)	*p*-Value	HR (95% CI)	*p*-Value
Presence of periodontal disease											
No	165,814	1660	0.89 (0.84, 0.94)	1 (reference)		1 (reference)		1 (reference)		1 (reference)	
Yes	43,806	500	1.16 (1.05, 1.27)	1.25 (1.13–1.38)	<0.001	1.23 (1.11–1.36)	<0.001	1.23 (1.11–1.36)	<0.001	1.21 (1.09–1.34)	<0.001
Number of lost teeth											
0	158,113	1658	0.97 (0.92, 1.02)	1 (reference)		1 (reference)		1 (reference)		1 (reference)	
1–5	43,291	402	0.81 (0.72, 0.90)	0.89 (0.80–0.99)	0.035	0.88 (0.79–1.08)	0.105	0.89 (0.81–1.09)	0.124	0.91 (0.84–1.07)	0.169
≥6	8216	100	1.20 (0.95, 1.46)	1.20 (0.98–1.47)	0.172	1.18 (0.97–1.43)	0.412	1.19 (0.97–1.42)	0.421	1.15 (0.95–1.41)	0.460
*p* for trend				0.781		0.051		0.053		0.059	
Tooth brushing frequency per day											
0–1	29,883	312	0.94 (0.83, 1.06)	1 (reference)		1 (reference)		1 (reference)		1 (reference)	
2	91,626	973	0.93 (0.87, 1.00)	1.00 (0.88–1.13)	0.979	1.00 (0.88–1.14)	0.989	1.00 (0.88–1.14)	0.978	1.00 (0.88–1.13)	0.955
≥3	88,111	875	0.96 (0.90, 1.03)	0.97 (0.86–1.11)	0.687	1.07 (0.94–1.22)	0.306	1.07 (0.94–1.22)	0.305	1.05 (0.92–1.20)	0.473
*p* for trend				0.603		0.183		0.184		0.335	
Dental visits for any reason											
No	116,980	1175	0.92 (0.87, 0.98)	1 (reference)		1 (reference)		1 (reference)		1 (reference)	
Yes	92,640	985	0.97 (0.91, 1.04)	1.07 (0.98–1.16)	0.124	1.08 (0.99–1.17)	0.084	1.08 (0.99–1.17)	0.090	1.01 (0.91–1.11)	0.858
Expert teeth scaling											
No	155,365	1589	0.93 (0.88, 0.98)	1 (reference)		1 (reference)		1 (reference)		1 (reference)	
Yes	54,255	571	0.99 (0.90, 1.08)	1.05 (0.95–1.15)	0.332	1.05 (0.97–1.24)	0.115	1.03 (0.92–1.24)	0.117	1.08 (0.96–1.21)	0.186

Notes: Event rates are reported in 10-year event rates (%). Multivariate model (1) was used to evaluate the association of each oral hygiene parameter with the occurrence of unruptured cerebral aneurysms with adjustments for age, sex, socioeconomic status, regular physical activity, alcohol intake, smoking status, body mass index (kg/m^2^), hypertension, diabetes mellitus, dyslipidemia, renal disease, and history of malignancy. Multivariate model (2) was used to evaluate the association of each oral hygiene parameter with the occurrence of unruptured cerebral aneurysms with adjustments for the variables in model 1, as well as systolic blood pressure, total cholesterol, fasting blood glucose level, aspartate aminotransferase, alanine aminotransferase, gamma-glutamyl transferase, and proteinuria. Multivariate model (3) was used to evaluate the association of each oral hygiene parameter with the occurrence of unruptured cerebral aneurysms with adjustments for the variables in model 2, as well as overall oral hygiene parameters (presence of periodontal disease, number of lost teeth, tooth brushing frequency per day, dental visits for any reason, and expert teeth scaling). Abbreviations: CI, confidence interval; HR, hazard ratio.

## Data Availability

Access to NHIS data can be requested from the website of the National Health Insurance Sharing Service (http://nhiss.nhis.or.kr/bd/ab/bdaba021eng.do, Access date: 28 August 2020). A completed application form, proposal for research, and approval document from the appropriate IRB must be submitted and reviewed by the NHIS inquiry committee for research support.

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
