# Peer review of "Association of Periodontal Disease with the Occurrence of Unruptured Cerebral Aneurysm among Adults in Korea: A Nationwide Population-Based Cohort Study"

_medicina, 2021, doi:10.3390/medicina57090910_

Round 1

Reviewer 1 Report

Dear authors now the critical parts that we evaluated are revised and improved. We consider the manuscript ready to be published

Author Response

To Reviewer 1

Comment 1. Dear authors now the critical parts that we evaluated are revised and improved. We consider the manuscript ready to be published.

Answer 1. We appreciate your comment.

Reviewer 2 Report

 I understand that this article was evaluated the association between intracranial aneurysm and the prevalence of periodontal disease in a field survey of Korean people.

 The relationship between cardiovascular disease and periodontal disease, was frequently discussed and its mechanism is being studied. There are relatively many studies on abdominal aortic aneurysms, but few studies on the relationship between cerebral aneurysms and periodontal disease.

I think, this is a meaningful paper in that it shows that periodontal disease increases the risk of cerebral aneurysms in Asians. I hope, if you will be clear whether or not there are differences depending on race, this paper will be more meaningful. In the future, I would like you to continue to consider the severity of periodontal disease and the number of years of history.

Author Response

We appreciate this insightful and critical comment. 

This manuscript is a resubmission of an earlier submission. The following is a list of the peer review reports and author responses from that submission.

Round 1

Reviewer 1 Report

I would like to thanks the authors for providing a well written and concise manuscript

I would have some remarks:

Major comments

-There is an obvious selection bias in considering patients that have a dental examination only. this is confirmed by the  good health level of patients included.

  • Overall there is a paucity of events with a low aneurysm rate that decrease the significance
  • With the third model the adition of  the dental variables takes the confounding factors better into account but there is also a risk of over-adjustment with so many variables.
  •  

Minor comments

-The Presentation of table 1 should be improved
- Authors shoud add a flow chart of inclusion

- there are some redundancies in the description of the models between material and results
- there are Abbreviations missing under the table

Author Response

We appreciate the reviewer’s comment. We revised manuscript (medicina-1206639) as editor’s comment and review’s comment. We wish that our findings contribute oral health research by appearing in the prestigious journal Medicina. We wish to thank you for consideration of our manuscript. 

Reviewer 2 Report

I consider this research very interesting.

About the methods i consider inappropriate the generalized "oral brushing" without any concern about the use of interdental devices, like oral irrigators. Please consider a new article published in Applied science, about oral hygiene maintenance in periodontal patients. ("Rough Dental Implant Surfaces and Peri-Implantitis: Role of Phase-Contrast Microscopy, Laser Protocols, and Modified Home Oral Hygiene in Maintenance. A 10-Year Retrospective Study" Appl. Sci. 2021, 11, 4985. https://doi.org/10.3390/app11114985

Please refer to this article and try to specify better the chapter 

Author Response

(The authors gave the same response as above.)

Round 2

Reviewer 2 Report

Even if the oral hygiene protocols aree not well described, I consider that the authors did an amazing research, able to show how periodontal disease is involved in systemic diseases. In effect this paper could help to detect early these important pathologies, when PD is evaluated.